# The effects of indoor temperature and humidity on local transmission of COVID-19 and how it relates to global trends

Han June Park[1], Sung-Gwang Lee[1], Jeong Suk Oh[2], Minhyuk Nam[3], Steven Barrett[2]*, Soohyung Lee[4]*, Wontae Hwang[1,5]*

**1** Department of Mechanical Engineering, Seoul National University, Seoul, South Korea, **2** Department of Aeronautics and Astronautics, Massachusetts Institute of Technology, Cambridge, Massachusetts, United States of America, **3** Department of Economics, Sogang University, Seoul, South Korea, **4** Graduate School of International Studies, Seoul National University, Seoul, South Korea, **5** Institute of Advanced Machines and Design, Seoul National University, Seoul, South Korea

☯ These authors contributed equally to this work.

* wthwang@snu.ac.kr (WH); soohlee@snu.ac.kr (SL); sbarrett@mit.edu (SB)

**Data Availability Statement:** All relevant data are within the paper and its Supporting Information files.

## Abstract

During the COVID-19 pandemic, analyses on global data have not reached unanimous consensus on whether warmer and humid weather curbs the spread of the SARS-CoV-2 virus. We conjectured that this lack of consensus is due to the discrepancy between global environmental data such as temperature and humidity being collected outdoors, while most infections have been reported to occur indoors, where conditions can be different. Thus, we have methodologically investigated the effect of temperature and relative humidity on the spread of expired respiratory droplets from the mouth, which are assumed to be the main cause of most short-range infections. Calculating the trajectory of individual droplets using an experimentally validated evaporation model, the final height and distance of the evaporated droplets is obtained, and then correlated with global COVID-19 spread. Increase in indoor humidity is associated with reduction in COVID-19 spread, while temperature has no statistically significant effect.

## Introduction

Ever since the World Health Organization (WHO) proclaimed the Severe Acute Respiratory Syndrome (SARS) Coronavirus (namely SARS-CoV-2 or COVID-19) as a pandemic in March 2020, there has been much debate regarding the transmission mechanisms of COVID-19. Transmission mechanisms of respiratory viruses such as influenza A/B, rhinovirus, and adenovirus, and their connections to climate effects such as temperature and humidity have been analysed in the past [1–5]. Recently, there have been several attempts to correlate temperature and humidity with the seasonality of COVID-19 infection rate [6–12]. However, temperature and humidity are not the only factors that affect seasonal virus infection rate. Other 'natural' factors such as human immune function and ultraviolet radiation and can play a role, and 'social' factors such as mobility patterns and school semester need to be additionally

**Funding:** This work was supported by Seoul National University Research Grant in 2020 and supported under the framework of international cooperation program managed by the National Research Foundation of Korea (2020K2A9A1A01096358, FY2020). The funders had no role in study design, data collection and analysis, decision to publish, or preparation of the manuscript.

**Competing interests:** The authors have declared that no competing interests exist.

considered [3]. It should be noted that statistical analyses often lack consensus due to the fact that researchers use different sample countries, time periods, as well as statistical models. Additionally, climate data are recorded outdoors, whereas most virus transmission is considered to occur indoors, which can have different conditions than the outside environment due to heating, ventilation, and air-conditioning (HVAC). This is affected by the socioeconomic status of the sample population and geometric location of the country as well. Therefore, no clear statistical pattern has yet been provided connecting global climate and infection rate statistics with local indoor virus transmission mechanisms.

Regarding virus transmission mechanisms, since COVID-19 transmission is mainly considered to occur through virus containing droplets expelled from an infected person's mouth, the fluid dynamics of airborne droplets and how it is directly affected by temperature and humidity need to be elucidated. However, until now most droplet dynamics simulations or analyses have been conducted based on existing evaporation models, which have not been extensively validated at various temperature and humidity conditions [13–21].

In the present work, we conduct a detailed analysis on how temperature and relative humidity (RH) affect the propagation of COVID-19. First, cross-country infection statistics are presented using a model that was built based on multivariate ordinary least squared regressions including extensive controls (e.g., air pollutants, country fixed effects and country-specific time trends). The estimated effects of RH and temperature across countries were compared to assess whether these two explanatory variables show robust relationships with COVID-19 spread. Next, the effects of temperature and RH on the physical transport of respiratory droplets expired from the mouth are investigated. A theoretical model of droplet evaporation is experimentally validated at various temperatures and RH. Based on this model, computational fluid dynamics (CFD) simulations were conducted, coupled with dynamic modelling, to analyse the transport of droplets in an indoor environment. Finally, we compare the simulation data with cross-country statistical results to predict the effects of temperature and RH on COVID-19 spread.

## Statistical methodology

### Data

We compiled a dataset that included confirmed COVID-19 cases, weather and pollution information, and government response toward COVID-19.

The COVID-19 Open Data provided by the Google Cloud Platform [22] contains daily COVID-19 epidemiological data, as well as daily weather and geographical data from 232 countries, in detailed spatial resolution up to the county or municipality level. We use this data to compute monthly new confirmed cases within a given location, as of the last day of each month. The weather data includes daily average temperature, relative humidity, and total precipitation. We take a simple average by month for each location. Geographical data includes latitude and longitude of the centroid for each region.

Pollution information is from the CAMS near-real-time surface-level data provided by the European Centre for Medium-Range Weather Forecasts (ECMWF). The dataset reports concentration levels for air pollutants such as carbon monoxide (CO), nitrogen monoxide (NO), nitrogen dioxide ($NO_2$), sulfur dioxide ($SO_2$), and particulate matter with aerodynamic diameter $\leq 2.5$ μm ($PM_{2.5}$) and $\leq 10$ μm ($PM_{10}$) every 6 to 12 hours (depending on the type of pollutant), on a 0.4˚ x 0.4˚ latitude and longitude grid. The unit of air pollutant concentration level (except particulate matter) is $g/m^2$ (and $pg/m^2$ for NO), because the dataset provides the total amount of atmospheric pollutants in a squared meter from the earth's surface to the top of the atmosphere, while particulate matter is measured in μg/m³. For each location, we use

the readings from the closest observation for the given coordinates, based on the geography information from the COVID-19 Open Data. To construct monthly air pollutant levels, we take the simple average from hourly data.

We obtain overall government response toward COVID-19 from the Coronavirus Government Response Tracker (OxCGRT), provided by the Blavatnik School of Government at the University of Oxford. For 177 countries, the dataset tracks 18 government policy responses on a daily basis in three different policy areas including containment (e.g., restriction in movement, closure of workplace or school), health system (e.g., accessibility of COVID-19 testing, amount of vaccine investment), and economic support (e.g., income support, debt relief to citizens). The dataset then provides a single index number for each area, normalized between 0 and 100. Most of the observations are measured at the country level (except several countries such as the U.S., Canada, UK, and Brazil), and we exploit the most detailed information available for the given location.

## Sample

Our data spans from January 1st, 2020, when the Chinese government first officially reported confirmed COVID-19 cases to the WHO, till February 10th, 2021 which was the latest data available when we conducted the analysis. Using this data, we take simple averages to construct monthly level data. By merging the three data sources described above, we constructed our sample including 1,085 locations (174 countries), while 961 locations are states or provinces from 50 countries. Note that among the 232 countries in the COVID-19 Open Data, 177 countries are included in the OxCGRT. Among them, 3 countries did not report positive cases during our sample period, and were thus excluded from our analysis. (see S1 File for details and a summary of the statistics).

## Statistical modelling

It may be worth explaining about the four features of our regression model. First, we use the natural logarithm of $Y_{l,c,t}$ as the dependent variable, which includes the numbers of newly infected and cumulative COVID-19 patients at location $l$ in country $c$ at the end of the given month $t$. We use the logarithmic transformation for two reasons. One is the fact that the difference in logs can be used to approximate proportionate changes. For example, let $y_0$ and $y_1$ be positive values. Then it can be shown that $\log(y_1) - \log(y_0) \approx (y_1 - y_0)/y_0 = \Delta y/y_0$ for small changes in $y$ (see details in Wooldridge 2009 [23]). Therefore, our regression model implies that a small increase in an explanatory variable (e.g., RH) would account for the estimated coefficient times 100 in percent change in the number of new COVID-19 cases (e.g., $100\gamma$). Moreover, the log transformation can reduce the effect of outliers in the data.

Second, we examine the number of newly infected patients, not the basic reproduction ratio R0, which is the expected number of secondary infections produced by a single infection during one's entire infectious period in a completely susceptible population. We made this choice because we did not want to introduce modelling assumptions on how the COVID-19 virus may spread, in order to estimate R0. Furthermore, the unit of time is month in our data, and thus the assumption to estimate R0 may not be valid in our setting. More specifically, popular models estimating R0 and susceptible–infectious–recovered (SIR) models assume that R0 is entirely accounted for by past R0, not by other explanatory variables [22, 24]. This assumption may be valid if a high frequency (e.g., hourly, daily) dataset is being examined, but unlikely if the time span is wide (e.g., month), because several factors affecting people's behaviors (e.g., lockdown) may vary across time.

Third, we used an extensive list of explanatory variables in our regression, including both country fixed effects and a country-specific linear time trend in our model. These two types of variables account for factors affecting the country-level evolution of COVID-19 spread, such as the frequency of holidays and degree of compliance that could affect outdoor activities and willingness to follow any lockdown orders. In addition, we included monthly average precipitation (mm), atmospheric pollutant concentration levels of CO, NO, $NO_2$, $SO_2$, $PM_{2.5}$ and $PM_{10}$, and three variables provided by the University of Oxford that capture government COVID-19 policies: degree of strictness on closure/containment (0: least strict, 100: most strict), coverage of health resources for contact tracing and vaccine procurement (0: no coverage, 100: highest coverage), and financial support to alleviate economic shock (0: no support, 100: highest support). Furthermore, we created discrete variables whose value indicates a specific month in a year (2020 or 2021), located in either the northern or southern hemisphere (noted by month x year x hemisphere fixed effects). In total, we used 385 explanatory variables (aside from temperature and RH), which include 174 country-specific intercepts, 174 country-specific linear time trends, and 37 other explanatory variables. In addition, we examined various alternative assumptions in the regression model (e.g., possible non-linear impact of pollutants), but found that our results were quantitatively stable. See Section 3.3 in the S1 File.

Fourth, we would like to note that the number of locations in a country greatly varies in our data. To address this asymmetry, we calculate the weight of a location as the inverse of the total number of locations in the corresponding country and apply these weights in our estimation, so that all countries have the same weight.

## Results

### Cross-country econometric analysis

We start by presenting the statistical pattern of COVID-19 spread due to temperature and humidity. Although various studies have examined such patterns so far, our aim is to extract a more stable statistical pattern using two methods. First, the data coverage is expanded across a relatively long time span and many countries [9, 11, 24–28]. Second, an extensive list of explanatory factors that could account for COVID-19 spread [9, 24, 26, 27] is included.

Our data includes monthly information of 1,085 locations across 174 countries, from January 2020 to February 2021. We construct and estimate an ordinary least squares (OLS) model:

$$\log\left(Y_{l,c,t}\right) = \alpha_c + \tilde{\alpha}_c t + \beta \cdot T_{l,c,t} + \gamma \cdot RH_{l,c,t} + \sum_k \Theta^k X_{l,c,t}^k + \varepsilon_{l,c,t} \tag{1}$$

where subscripts $l$, c, and $t$ refer to location, country, and time (defined by month and calendar year). Variable $\log(Y_{l,c,t})$ represents the natural logarithm of the number of newly infected COVID-19 patients at the end of the corresponding month at location $l$ in country $c$. Coefficients $\alpha_c$ and $\tilde{\alpha}_c$ capture the country-specific intercept (commonly referred to as country fixed effects) and country-specific linear time trend. Variables $T_{l,c,t}$ and $RH_{l,c,t}$ refer to monthly average temperature (˚C) and RH (%), respectively. $X_{l,c,t}^k$ is the $k^{\text{th}}$ element of the vector $X_{l,c,t}$, which includes 37 other explanatory variables that may affect COVID-19 spread including precipitation, air pollutant levels, and government-wise policy responses to COVID-19. Variable $\varepsilon_{l,c,t}$ captures residual shock unexplained by all of these explanatory variables, and it is assumed to follow a distribution with zero mean and finite variance, and is independent from other residual shocks. Note that our results remain stable regardless of the assumptions we use in our regression analysis. See details in Section 3.3 of the S1 File.

Table 1 presents the effect of temperature and RH on COVID-19 infection. Baseline results are presented in column (1). For a given location and month, 1˚C higher temperature is

**Table 1. Effect of temperature and relative humidity on the logarithm of monthly new COVID-19 cases and cumulative cases across 174 countries from Jan. 2020 to Feb. 2021.**

| Period | 2020/1 to 2021/2 | 2020/2 to 2021/2 | 2020/3 | 2021/2 | 2020/3 | 2021/2 |
|---|---|---|---|---|---|---|
| Model | Baseline | Baseline + Past cases | OLS | OLS | Weighted OLS + Group FE | Weighted OLS + Group FE |
| | (1) | (2) | (3) | (4) | (5) | (6) |
| **A. log(new cases)** | | | | | | |
| Temperature | -0.027*** | -0.038*** | -0.021 | -0.008 | 0.044*** | -0.083*** |
| | (0.000) | (0.000) | (0.014) | (0.013) | (0.000) | (0.001) |
| | [0.000] | [0.000] | [0.140] | [0.570] | [0.000] | [0.000] |
| RH | -0.009*** | -0.011*** | 0.007 | 0.012 | -0.017*** | -0.016*** |
| | (0.000) | (0.000) | (0.010) | (0.010) | (0.000) | (0.000) |
| | [0.000] | [0.000] | [0.476] | [0.256] | [0.000] | [0.000] |
| R-sq | 0.827 | 0.867 | 0.177 | 0.162 | 0.450 | 0.356 |
| **B log(cum. cases)** | | | | | | |
| Temperature | -0.008*** | -0.012*** | -0.045*** | -0.013 | 0.035*** | -0.084*** |
| | (0.000) | (0.000) | (0.010) | (0.013) | (0.000) | (0.001) |
| | [0.000] | [0.000] | [0.000] | [0.323] | [0.000] | [0.000] |
| RH | -0.002*** | -0.001*** | -0.009 | 0.005 | -0.016*** | -0.022*** |
| | (0.000) | (0.000) | (0.007) | (0.010) | (0.000) | (0.000) |
| | [0.000] | [0.000] | [0.231] | [0.599] | [0.000] | [0.000] |
| R-sq | 0.903 | 0.964 | 0.242 | 0.135 | 0.432 | 0.352 |

_Notes_: The dependent variable of each panel A and B are the natural logarithm of new and cumulative infection cases as of the last day of each month. Columns (1)–(2) exploit panel data from January or February 2020 to February 2021, while columns (3)—(6) use cross-sectional data as of March 2020 and February 2021. Columns (1)–(2) include country fixed effects, month x calendar year x hemisphere fixed effects, and country-specific linear time trends, while columns (3)–(4) do not include any of these effects. Columns (5)–(6) include fixed effects of 8 country groups, which consist of geographically neighboring countries. Columns (1)–(2) and (5)–(6) are weighted by the reciprocal number of monthly observations for each country. Standard errors and p-values are in parentheses and square brackets, respectively. The p-values are denoted as * for p < 0.10, ** for p < 0.05, and *** for p < 0.01.

associated with 0.027 reduction in log ($Y_{l,c,t}$). In other words, 1˚C higher temperature suggests, on average, a 2.7% reduction in the number of new COVID-19 patients. Likewise, 1% pt. increase in RH is associated with a 0.9% reduction. These results are both statistically significant at the 1% level (p-values are less than 0.001 as denoted by the *** symbol). The bottom panel B of column (1) shows that the statistical patterns observed above are stable even for the number of cumulative COVID-19 patients. The coefficients of T and RH are both negative and statistically significant, similar to the case of new patients. Column (2) adopts the baseline statistical model in column (1), but additionally includes the cumulative number of previous cases as an additional explanatory variable. This allows for the possibility that new COVID-19 infection can be history dependent. It can be seen that the results are still comparable to the baseline results, indicating that the main results are stable. We find that the estimated coefficients for temperature and RH remain comparable across various regression specifications, even when including additional explanatory variables such as the difference in daily maximum and minimum temperature and quadratic terms for RH. We further conducted subgroup analysis on seasons of the year and region, and included the possibility of non-linear effects of temperature and humidity on COVID-19 transmission. See details in the S1 File (section 3.3).

The rest of Table 1 adopts statistical models in line with existing studies [9, 26, 27], and shows that a simple cross-sectional analysis (at a certain point in time) with a limited set of explanatory variables may be inappropriate to derive stable statistical patterns. For example, we select one month–either March 2020 (when COVID-19 was declared a pandemic), or

February 2021 (the final month of our data set)–and estimate a statistical model excluding country fixed effects and country-specific linear time trend. Columns (3) and (4) use a simple OLS, while columns (5) and (6) are weighted by the reciprocal number of monthly observations of each country to ensure comparability across countries. In columns (5) and (6), we also classify countries into 8 groups based on their geographical location and include group-specific intercepts, in order to compensate for the lack of country fixed effects. The reported estimates for columns (3)–(6) vary substantially in terms of their signs and statistical significance, suggesting that the lack of consensus on the role of temperature and humidity in existing studies is due to the difference in statistical models and sample choices. In addition to the fact that our statistical model delivers stable patterns between temperature, RH, and COVID-19 spread, the model's explanatory power is substantially better than the statistical methods used in existing studies. Specifically, R-squared utilizing our baseline model in columns (1)–(2) varies between 0.8–0.96, while those reported in columns (3)–(6) are less than 0.5.

## Droplet evaporation model

In order to correlate global statistics with local virus transmission, we first assume that the main transmission mechanism is through virus containing droplets ejected from the mouth. These droplets evaporate as they move in the air. The evaporation rate varies with temperature and relative humidity, and this changes the trajectory of the droplets. Therefore, in order to analyze the aerodynamic behavior of droplets, an appropriate evaporation model is required. Both actual saliva and water were considered for the droplets. Although the composition of saliva varies depending on a person's health condition, the general physical properties of saliva (i.e. density, surface tension, viscosity) differ from water within 0.5% [29, 30]. Therefore, we set up the evaporation model to use water. The D square law, which has been used for droplet evaporation in many previous studies, is also employed here [31–33]. Droplet mass transfer is expressed as follows:

$$-\frac{\mathrm{d}m_d}{\mathrm{d}t} = \frac{C_{Diff}}{d} A\rho_g Sh \ln(1 + \mathrm{B_m}) \tag{2}$$

where $m_d$ is droplet mass, $C_{Diff}$ is the air diffusion coefficient, $d$ is droplet diameter, $A$ is droplet surface area, $\rho_g$ is gas density, and $B_m = (Y_s - Y_g)/(1 - Y_s)$ is the Spalding mass transfer number, where $Y_s$ is vapor mass fraction at the droplet surface, and $Y_g$ is vapor mass fraction in the ambient gas. The subscripts 'd' and 'g' denote droplet and gas, respectively. The Sherwood number $Sh$ (the ratio of convective and diffusive mass transfer rates) can be expressed as [34]:

$$Sh = 2 + 0.6Re_d^{0.5}Sc^{1/3} \tag{3}$$

where the droplet Reynolds number is $Re_d = \rho_g d|\overrightarrow{V_d} - \overrightarrow{V_g}|/\mu$ and the Schmidt number is $Sc = \mu/(\rho_g D_g)$. $\overrightarrow{V_d}$ and $\overrightarrow{V_g}$ are droplet velocity and ambient gas velocity, respectively, and $\mu$ is gas dynamic viscosity. Assuming that the Lewis number (the ratio of Schmidt number and Prandtl number) is unity, $B_m$ becomes the same as $B_h$ (Spalding heat transfer number), and is expressed as follows [31]:

$$B_h = \frac{C_g(T_g - T_s)}{L_s} \tag{4}$$

where $C_g$ is specific heat of gas, $T_g$ and $T_s$ are ambient gas temperature and droplet surface temperature, respectively, and $L_s$ is latent heat. Assuming that the surface of the droplet where evaporation occurs is at saturation, $T_s$ becomes the dew point and can be calculated using the

Magnus formula [35]:

$$p_v(T_g) = p_{v,saturated}(T_s) \tag{5}$$

$$p_{v,saturated}(T_s) = C_1 exp\left(\frac{C_2 T_s}{C_3 + T_s}\right) \tag{6}$$

$$T_s = \frac{C_3 \ln\left(\frac{p_v(T_d)}{C_1}\right)}{C_2 - \ln\left(\frac{p_v(T_d)}{C_1}\right)} \tag{7}$$

where $p_v(T_g)$ is partial vapor pressure of the ambient gas, $p_{v,saturated}(T_s)$ is saturated vapor pressure at the droplet surface, $C_1$ is 610.94 Pa, $C_2$ is 17.625, and $C_3$ is 243.04°C [36].

It is assumed that the droplet concentration is dilute, and the size is small enough to ignore the interaction between them. The force balance equation considers body force, drag force, and buoyancy. Because the droplet size is on the order of micrometers, Stokes drag and Cunningham correction factor $C_c$ were used, and is expressed as follows, coupled with Eq (2):

$$\frac{d\overrightarrow{V_d}}{dt} = \overrightarrow{g}\left(1 - \frac{\rho_g}{\rho_d}\right) - \frac{9}{8}\frac{\mu}{d^2 \rho_d C_c}\left(\overrightarrow{V_d} - \overrightarrow{V_g}\right) \tag{8}$$

where $\overrightarrow{g}$ is gravitational acceleration.

## Experimental validation of droplet evaporation model

The droplet evaporation model needs to be experimentally validated in terms of temperature and humidity. There have been few studies in the past that have thoroughly investigated this in an environment with a wide range of independent control of both temperature and humidity. Thus, we conducted droplet evaporation experiments in an environmental chamber which can separately control temperature (between 20–30°C) and relative humidity (between 20–80%), as shown in Fig 1(A). A humidifier generates many micron sized droplets, which coalesce into approximately a 200–300 μm droplet within an acoustic levitator [37], which holds the larger droplet in air using a 40 kHz stationary wave generated by 72 transducers (as depicted in Fig 1B). The droplet is illuminated by a high-power 532 nm LED and imaged using a PHANTOM V2640 high-speed camera. Using a 200 mm lens and teleconverter (2x, 3x), a 7.5 x 7 mm$^2$ region was imaged at a resolution of 2K x 2K pixels. The camera exposure time is 1 μs and frame rate is set to 100 frames per second.

The environmental chamber internal temperature and RH were controlled to be within ±1°C and ±1%, respectively. A handheld temperature and humidity meter was placed near the acoustic levitator to provide additional local measurements near the evaporating droplet. When conducting the experiment, the chamber was temporarily turned off while measurements were taken, to eliminate any internal ventilation flow. The temperature and RH were controlled between 20–30°C and 40–80%, respectively, to represent indoor closed room conditions where most virus transmission occurs.

Droplet evaporation experiments were conducted to validate the evaporation model in indoor environment conditions. A sample result is shown in Fig 2 (see S1 File for additional experimental results). The droplets were imaged via shadowgraphy (see S2 File), as illustrated in Fig 2(A), and the diameter change with time was measured (see S1 File). Most initial droplet diameters were in the range of 80 to 200 μm. The droplets moved a bit within the acoustic levitator, and thus the additional convective mass transfer due to this "streaming" effect altered

(a)

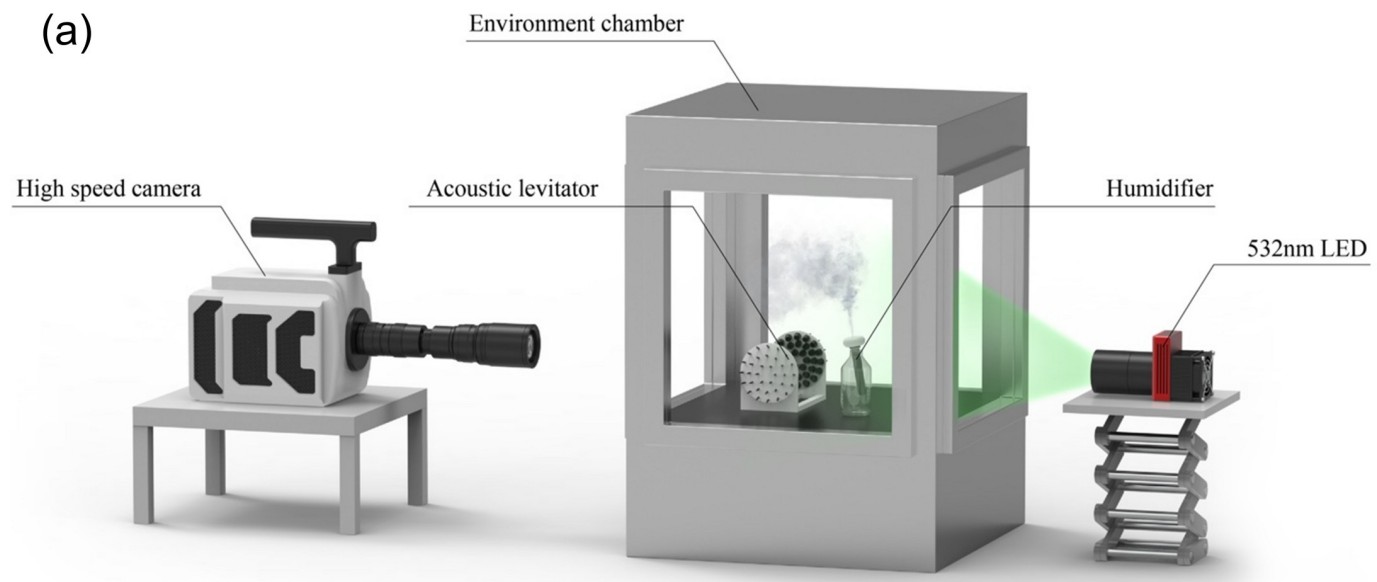

(b)

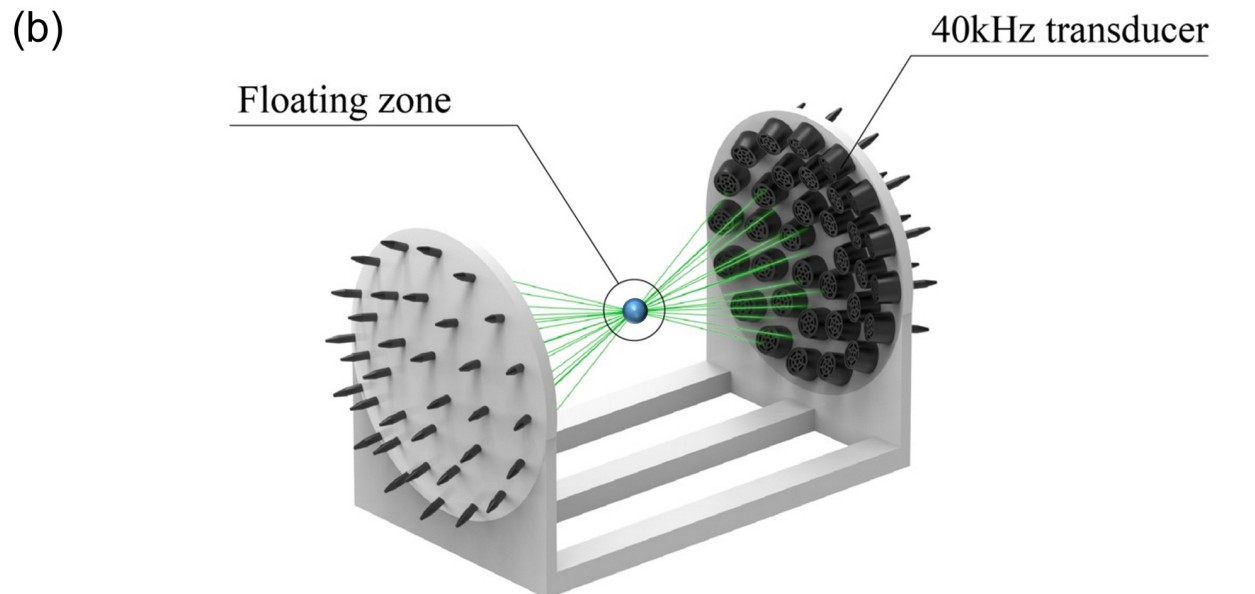

**Fig 1. Schematic of experimental setup.** (a) Droplet evaporation experimental setup, (b) schematic of acoustic levitator.

the Sherwood number *Sh* during droplet evaporation [38, 39]. Therefore, we made corrections for *Sh* in the evaporation model (see S1 File). As shown in Fig 2(B), we measured the droplet size change over time for an initial droplet size $D_0$ of 200 μm, at constant temperature and RH. The non-dimensional size is defined as $D^* = D/D_0$ and non-dimensional time is $t^* = t/t_0$, where $t_0 = (D_0^2/C_{Diff})(\rho_l/\rho_g)$, which is derived from the integral of Eq (2). $C_{Diff}$ is the air diffusion coefficient, $\rho_l$ is liquid (i.e., water) density, and $\rho_g$ is gas density. The slope of the curves

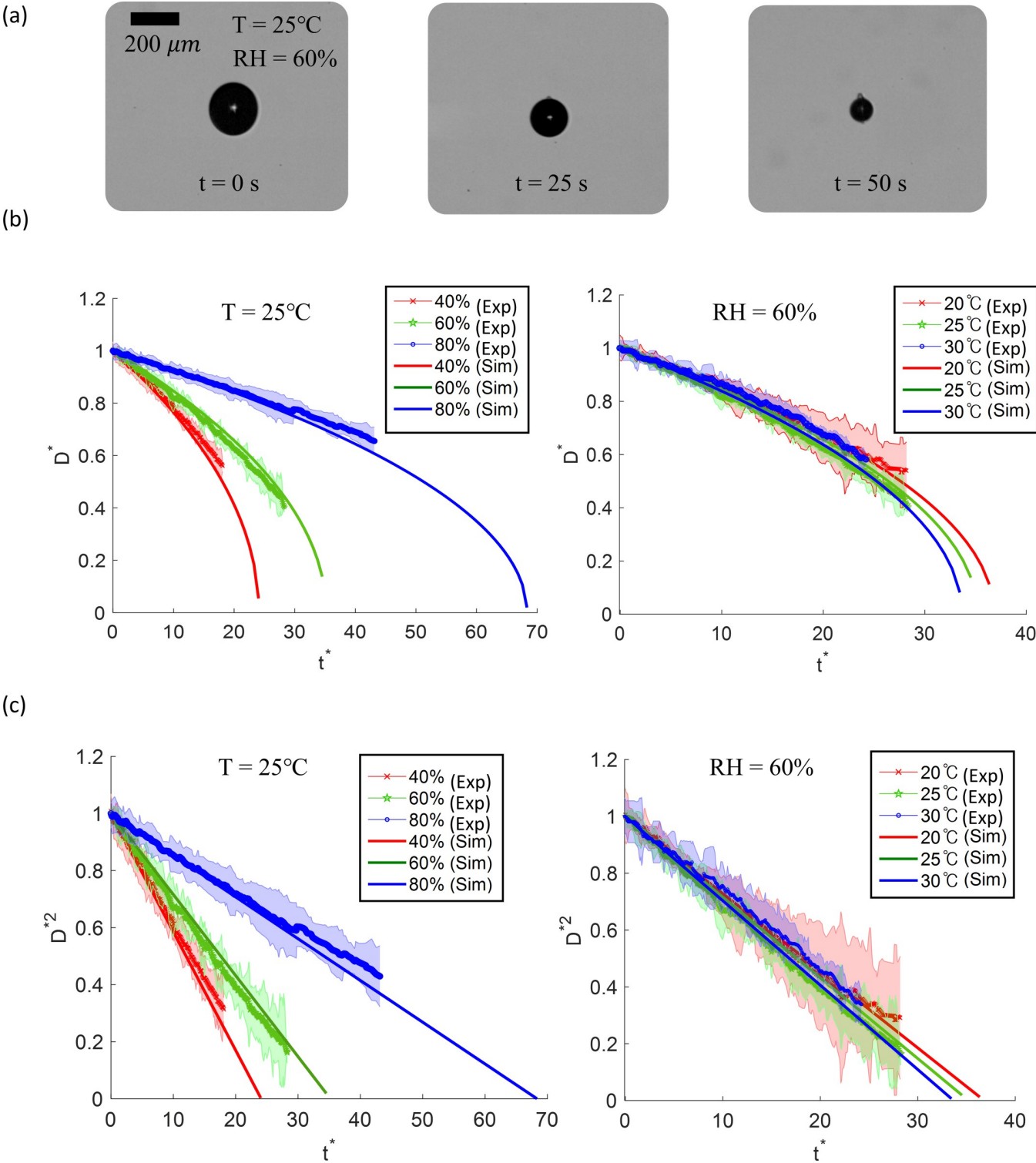

**Fig 2. Experimental and modelling results of evaporating droplet.** (a) Microdroplet evaporating at 25˚C and 60% relative humidity, (b) comparisons of diameter of droplet evaporation model and experiment, (c) comparisons of diameters squared of droplet evaporation model and experiment.

corresponds to the evaporation rate. $D^*$ decreases nonlinearly with $t^*$, with faster evaporation at small diameters, as shown in Fig 2(B). However, $D^{*2}$ decays linearly with $t^*$, as shown in Fig 2(C).

The experimental results match the theoretical model fairly well. When RH increases from 40 to 80%, the evaporation rate decreases (in a non-linear fashion) at constant temperature. On the other hand, when temperature increases from 20 to 30°C, the theoretical evaporation rate only slightly increases at constant RH, while the experimental trends cannot be differentiated within experimental uncertainty (at 95% confidence level from five tests). Although the temperature variation is not large, it is representative of indoor environments. These results indicate that the droplet evaporation rate is much more sensitive to relative humidity than temperature.

## Droplet transport modeling

Previous studies suggest that COVID-19 has two main paths of transmission, according to the U.S. Centers for Disease Control and Prevention (CDC) [40]. The first is transmission via respiratory secretions or droplets, either directly to a person or indirectly onto a surface. This is considered to be the general path, and occurs in the vicinity of the infected person. The second path is airborne transmission, where people are exposed to very small nuclei (aerosols) above a certain concentration level, either from normal breathing or talking. The droplet behaviour is different according to the transmission path, and this study examines the droplet behaviour of both paths.

Expiratory activities include breathing, talking, laughing, coughing, and sneezing. All activities are known to influence virus transmission. For instance, viral droplets can be spread by just talking and even breathing [41]. However, coughing and sneezing produce more droplets than the other activities [42], and the droplet propagation distance is also longer. This study focuses on coughing because it generally occurs more often than sneezing, and it is a typical symptom of COVID-19 [43].

The air discharged from a person's mouth due to a cough was simulated as an unsteady jet (typical flow profile is shown in Fig 3A) [43]. Factors influencing droplet behaviour include initial droplet size, transient jet initially carrying the droplet, buoyancy of the jet, and droplet evaporation rate. Ventilation at the ISO 7730 indoor thermal standard [44] was considered, such that a steady stream of 0.175 m/s ambient air was applied in the direction of the jet. This corresponds to the worst case ventilation scenario where the droplet will travel the farthest.

Droplet aerodynamic behaviour was calculated using two tools. First, the cough event was simulated as a transient jet ejected from the mouth at bodily conditions, entering the room at a certain temperature and RH (Fig 3B). The jet was calculated from a 2D transient analysis using ANSYS Fluent Reynolds averaged Navier-Stokes (RANS) CFD, which utilized the Shear Stress Transport (SST) turbulence model [45]. Next, droplet trajectories emanating from the jet were calculated using a Matlab in-house code. At every time step, the ambient flow velocity, temperature, and RH around the droplet were obtained, which enabled calculation of droplet evaporation and drag. Calculations were performed for a maximum of 18 seconds, and stopped when the droplet fell to the floor, or when its radius was less than 2 μm. Since there is a temperature difference between the warm jet ejected from the body and the relatively cooler ambient air, buoyancy effects were taken into account. The temperature drop of the jet was calculated using energy conservation. Relative humidity was calculated using a mixture of air and water vapor with a multicomponent model.

An initial droplet diameter ranges of 50 to 150 μm, which accounts for roughly 70% of the distribution of droplets generated by a cough (Fig 4A) [46], was used. Nine cases were

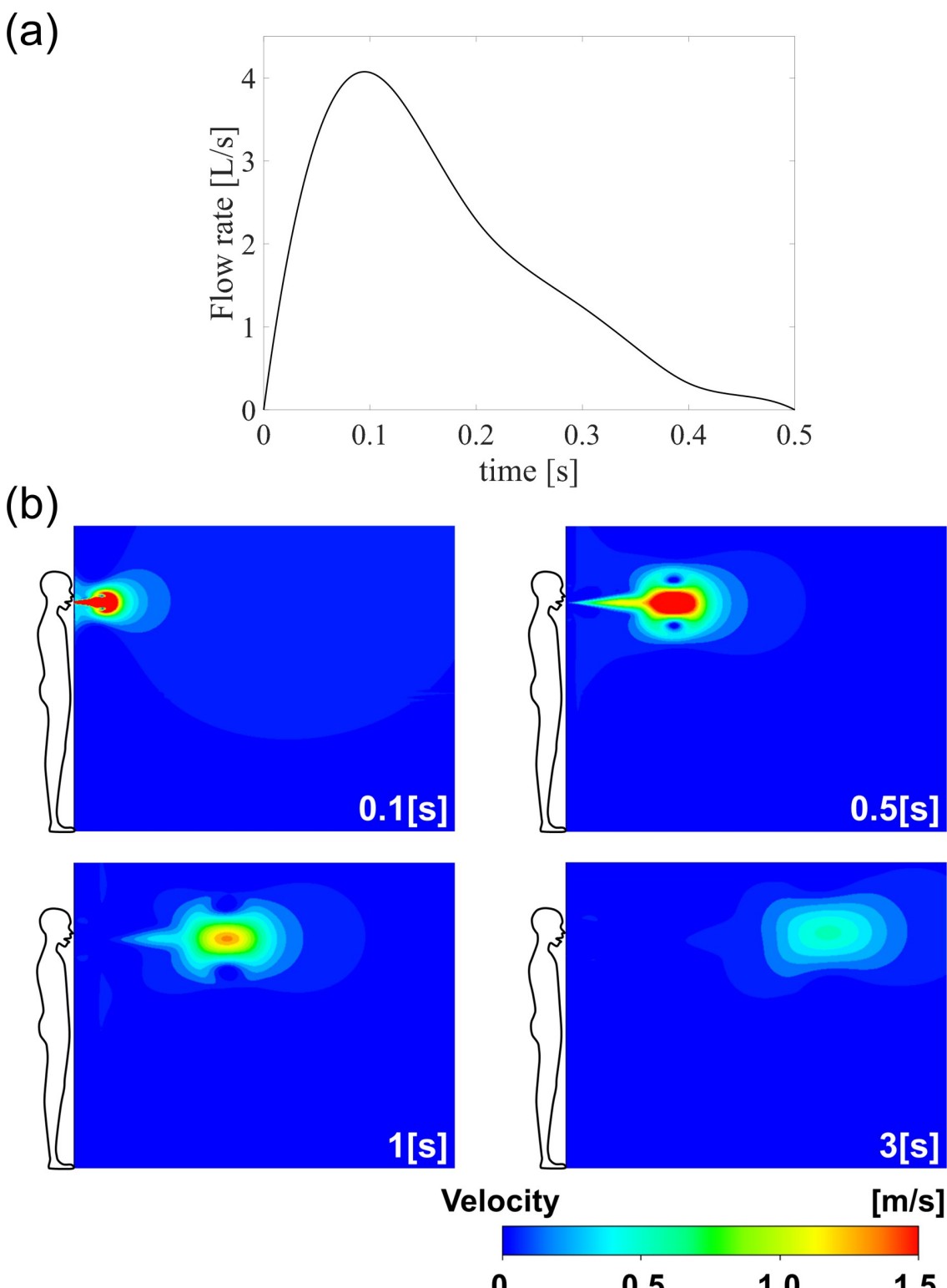

**Fig 3. Initial flow rate profile and transient flow field of jet from a cough.** (a) Initial flow rate of transient cough event, (b) behaviour of transient cough jet within indoor environment condition of 20˚C and 40% relative humidity.

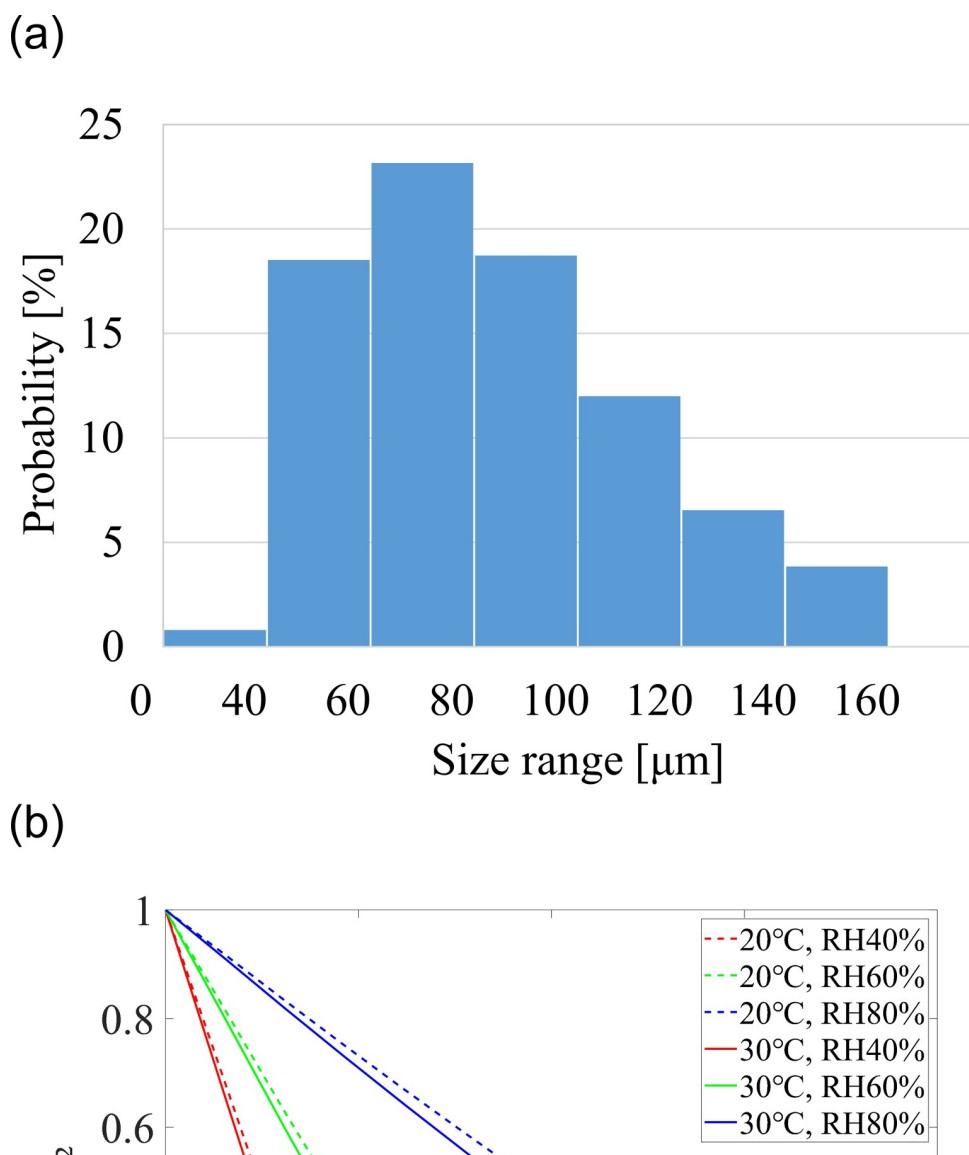

**Fig 4. Droplet size distribution and evaporation rate.** (a) Size distribution of initial cough droplets, (b) diameter change according to temperature and relative humidity for 70 μm initial droplet.

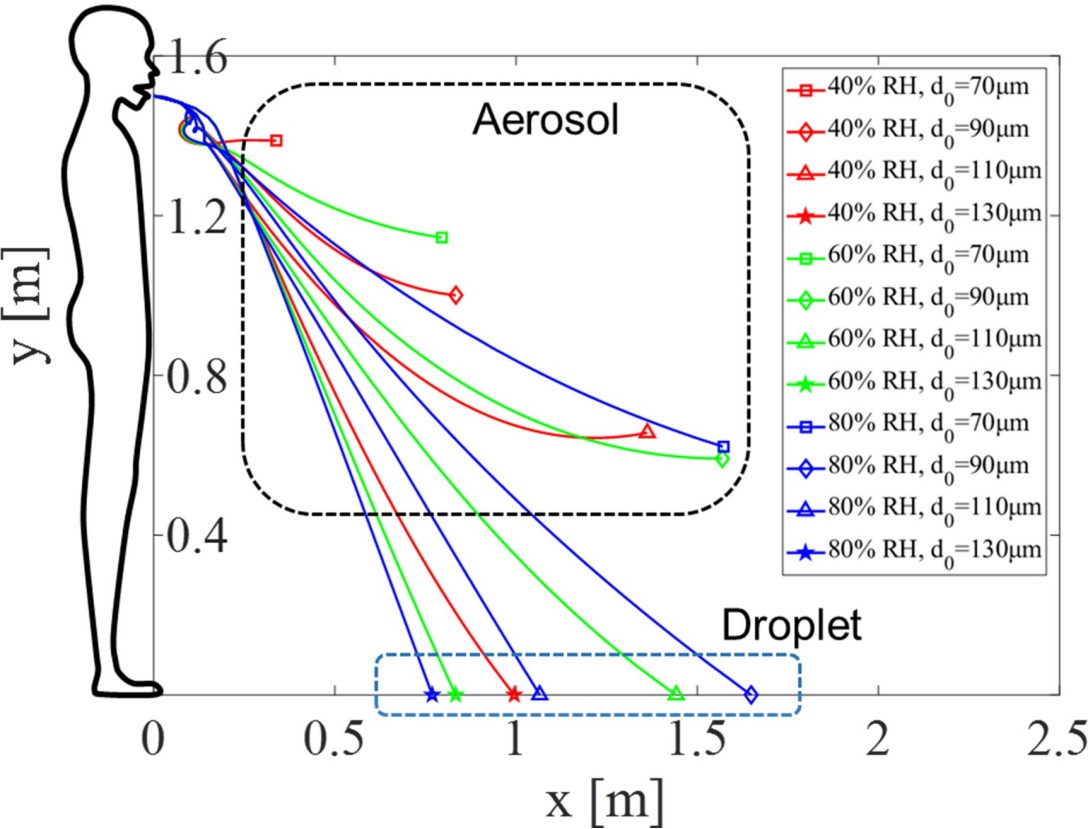

**Fig 5. Droplet trajectories according to ambient relative humidity and initial droplet size at a fixed room temperature of 20°C.**

examined using a combination of three different temperatures (20, 25, 30°C) and RH (40%, 60%, 80%), which were considered representative indoor conditions.

We first verified whether the evaporation model was properly applied within the CFD. For each case of temperature and RH, the droplet diameter variation along the trajectory was examined. Fig 4(B) shows how the droplet size changes over time for an initial 70 μm droplet. The trends are not completely linear as in the experimental results of Fig 2(C), because the ambient flow, temperature, and RH slightly change at every moment along the trajectory. Nonetheless, it can be seen that the CFD results of the cough jet also demonstrate that the evaporation rate is more sensitive to relative humidity than temperature.

Droplet trajectories were simulated next, assuming the average height of a person's mouth is roughly 1.5 m. A droplet is ejected from the mouth within the transient jet, initially assuming a relative humidity of 100%. Fig 5 shows trajectories according to ambient RH for initial droplet sizes ($d_0$) between 70 and 130 μm, at room temperature of 20°C.

When $d_0$ is large (e.g. 130 μm), the droplet does not completely evaporate and falls to the floor under all RH conditions, and with higher RH the droplets have a shorter travel distance. From Eqs (3)—(8), it can be seen that the evaporation rate decreases as the relative humidity increases, resulting in a droplet remaining larger in size. From the dynamic model Eq (8), if the droplet size is large the drag force becomes relatively small compared to the gravitational body force, such that the travel distance and fall time to the ground both become shorter. It is interesting to note that the travel distance is about 0.8–1.7 m in this case, which is shorter than

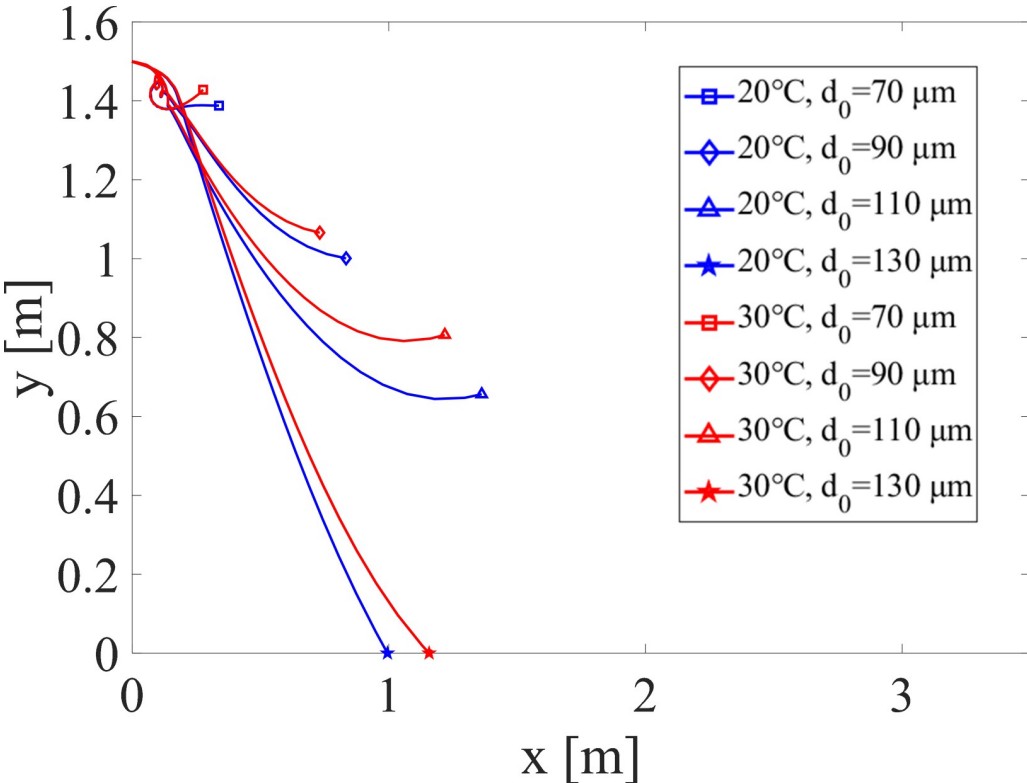

**Fig 6. Droplet trajectories according to temperature at 40% relative humidity.**

the 2 m (or 6 feet) that the WHO and many countries use as a social distancing guideline to prevent COVID-19 transmission.

When $d_0$ is small (e.g. 70 μm), the droplet evaporates in air and becomes an aerosol for all RH cases. With higher RH the droplet (before full evaporation) has spread farther and the final height is lower at the fully evaporated state. On the contrary, after full evaporation, the aerosol is likely to spread farther when RH is low because the advection starts at a higher height than a high RH case. According to previous studies, virus infection by aerosols can occur when one is exposed to a certain concentration over a certain period of time [40]. The higher the humidity, the aerosols are dispersed at a lower height compared to the source, resulting in a lower chance of inhalation since it is further away from the human respiratory tract. In addition, since the evaporation rate decreases with higher RH, the influence of the gravitational body force increases accordingly, and the droplets fall further to the ground. That is, although aerosols can cause infection, with higher humidity the final aerosol height is lower, likely resulting in a relatively safer condition.

In regards to temperature, the droplet evaporation rate is not as sensitive to this parameter compared to relative humidity. Fig 6 shows the droplet trajectory according to temperature when $d_0$ is 70 to 130 μm, at 40% RH which is representative of room conditions. The evaporation rate increases slightly with temperature, resulting in a lighter droplet. Therefore, the gravity force decreases and the droplet final height after evaporation is a little higher for the smaller droplets. For the largest $d_0$ of 130 μm, the droplet travels slightly farther at the higher temperature, before hitting the ground. However, compared to humidity, there is not a large difference between height and distance, indicating that temperature is a much less sensitive parameter in regards to droplet trajectory.

## Correlation between indoor temperature & RH with global COVID-19 infection

This section takes the simulation results of droplet transport and assesses the implications regarding COVID-19 infection. To do so, our statistical analysis is divided into three steps. First, we estimate an OLS regression to relate indoor temperature, RH, and initial droplet diameter to three outcome variables: time elapsed in the air (i.e., the duration of time from when the droplet was first released until it hits the ground) final distance travelled, and final height of the droplet. This allows us to predict these three outcome variables based on various combinations of temperature and RH. Second, using our COVID-19 dataset, we estimate the extent to which the number of global new COVID-19 cases is associated with the predicted time elapsed, final distance, and final height of the droplet. Third, we combine the results from the previous two steps and calculate the effect of indoor temperature and RH on droplet transport and how it affects global COVID-19 infections.

**Step 1: Correlating indoor temperature and RH to time elapsed, final distance, and final height.** We regress each of the three outcome variables (i.e., time elapsed, final distance, and final height) on temperature, RH, and initial droplet diameter. The data is divided into two groups. One includes droplets that eventually drop to the ground, and the other includes those that fully evaporate. Table 2 presents the results. For both groups of droplets,

**Table 2. Effect of droplet transport characteristics on newly infected cases.**

| | Dropped to the ground | | Fully evaporated | | |
|---|---|---|---|---|---|
| | Time Elapsed | Final Distance | Time Elapsed | Final Distance | Final Height |
| | (1) | (2) | (3) | (4) | (5) |
| **A. 1st Step** | | | | | |
| Temperature | 0.024 | 0.004 | -0.080 | -0.019* | -0.001 |
| | (0.028) | (0.005) | (0.055) | (0.011) | (0.011) |
| | [0.403] | [0.481] | [0.165] | [0.085] | [0.956] |
| RH | -0.033*** | -0.006*** | 0.194*** | 0.037*** | -0.011*** |
| | (0.008) | (0.001) | (0.016) | (0.003) | (0.003) |
| | [0.001] | [0.001] | [0.000] | [0.000] | [0.003] |
| R-sq | 0.955 | 0.886 | 0.892 | 0.928 | 0.661 |
| **B. 2nd Step** | | | | | |
| log(new cases) | 0.215*** | 1.217*** | -0.035*** | -0.186*** | 0.589*** |
| | (0.050) | (0.282) | (0.008) | (0.043) | (0.135) |
| | [0.000] | [0.000] | [0.000] | [0.000] | [0.000] |
| **C. Implication** | | | | | |
| ↑1% pt. in RH → log(new cases) | -0.00710*** | -0.00730*** | -0.00679*** | -0.00688*** | -0.00648*** |
| | (0.002) | (0.002) | (0.002) | (0.002) | (0.001) |
| | [0.000] | [0.000] | [0.000] | [0.000] | [0.000] |

**Notes:** In panel A, three droplet transport characteristics (i.e. time elapsed, final distance, and final height) from the CFD simulations are used as the dependent variable. All simulation results assume uniform ventilation flow in an indoor setting, and indicators of initial droplet size are also controlled. In panel B, the dependent variable is the natural logarithm of newly infected cases, which corresponds to the residual of Eq (1), excluding temperature and relative humidity. Dependent variables are explained by a single droplet transport characteristic such as time elapsed, final distance, and final height, which are predicted by estimates from panel A using converted indoor temperature and relative humidity. We use simulation data of droplets which hit the ground before they fully evaporate for columns (1)–(2), and droplets that fully evaporate for columns (3)–(5). All regressions are weighted by the reciprocal number of monthly observations for each country. Standard errors reported in parentheses are bootstrapped within the same countries 1000 times. P-values are in square brackets, and are denoted as * for $p < 0.10$, ** for $p < 0.05$, and *** for $p < 0.01$.

temperature has no statistically significant power (even at the 10% level) to account for the three outcome variables. However, RH shows a statistically significant trend at the 1% level. An increase in RH is negatively associated with time elapsed and final distance of droplets that drop to the ground, as can be seen in columns (1)—(2). Thus, when the indoor RH is higher by 1% pt., droplets fall to the ground 0.033 sec faster and travel 0.006 m less. For the fully evaporated droplets, higher RH increases time elapsed and significantly increases final distance, but slightly reduces final height, as evidenced in columns (3)—(5). In other words, with 1% pt. higher RH, the droplets take 0.194 sec longer to fully evaporate, travel 0.037 m more, and end up at a 0.011 m lower final height.

**Step 2: Correlating time elapsed, final distance, and final height to global COVID-19 infections.**   Using our baseline regression Eq (1), we calculate the difference between the actual and predicted dependent variable, where the prediction assumes no dependence on temperature, RH, and shock, by setting, $\gamma$, and $\varepsilon_{l,c,t}$ to be zero. This difference measures the logarithm of newly confirmed COVID-19 cases that are explained entirely by temperature, RH, and shock. We then treat this difference as the new dependent variable and examine its statistical relationship with the three outcomes of droplet transport: time elapsed, final distance, and final height. To do so, we first take each combination of temperature and RH in our COVID-19 infection dataset (measured at outdoor weather stations) and convert it to indoor temperature and RH (based on Nguyen et al., 2014 & Nguyen and Dockery 2016 –see S1 File for detail), because the droplet experiments and simulations were conducted in an indoor setting. Based on the converted indoor temperature and RH, we estimate how the three outcomes of droplet transport affect the new dependent variable [47, 48]. Note that we calculate the standard errors of each coefficient using the bootstrap method based on 1000 simulations to reflect estimation errors from each step.

The estimated coefficients are reported in the middle rows of Table 2. For example, column (1) implies that a unit (i.e., 1 sec) increase in time elapsed is associated with a 21.5% increase in the number of new COVID-19 infection cases, and this effect is statistically significant at the 1% level. Similarly, in column (2) a unit (i.e., 1 m) increase in final distance for droplets falling to the ground corresponds to a 122% increase in new cases. The estimated coefficients in columns (3)—(5) indicate that an increase in time elapsed and final distance for evaporated droplets reduces the number of new cases (due to the reduced concentration of small aerosols entering the human respiratory tract), while an increase in final height by 1 m significantly increases the number of new cases by 58.9%.

**Step 3: Connecting indoor temperature and RH to COVID-19 infection via droplet transport characteristics.**   The final step is to examine the role of indoor temperature and RH on COVID-19 infection through droplet transport, by connecting the results from the previous two steps. From our findings in step 1, RH has significant statistical relationship with all three outcomes of droplet transport, while temperature does not. Thus, we focus only on RH in this final step. The results are presented in the bottom rows of Table 2. Column (1) shows that a 1% pt. increase in indoor RH is associated with a 0.710% reduction in new global COVID-19 cases, due to 0.033 sec shorter time elapsed of falling droplets. Similarly, in columns (2) to (5), the effect of a 1% pt. RH increase results in 0.648% - 0.730% reduction of new cases for both falling and evaporating droplets, due to the change in droplet transport characteristics. These estimates, all statistically significant at the 1% level, are slightly smaller than our estimated result from Table 1 column (1), which captures the overall effects of relative humidity on new COVID-19 cases (i.e., -0.009***).

This change in number of new cases may appear insignificant at first glance. However, this is because a small 1% pt. increment in the independent variable RH was used. If we consider the fact that the standard deviation of indoor RH in our sample is 12.76% pts., one standard

deviation increase in indoor RH corresponds to a 9.1% reduction in new COVID-19 cases, which is quite substantial. If the indoor RH increases a bit more, for example an interquartile change from 46.4% (25[th] percentile of indoor RH in our sample) to 66.1% (75[th] percentile), the number of new cases decreases by 14%.

## Discussion

In this study, we examined the effect of temperature and relative humidity on respiratory droplet dynamics, and correlated this with COVID-19 transmission. The droplet evaporation model was experimentally validated in an environmental chamber representing indoor conditions. Droplet trajectories were simulated, and we analysed how the droplet dynamics vary with temperature and humidity. In this process, two conditions were considered. First, we used indoor conditions since most COVID-19 transmission is considered to occur indoors. In order to make the connection between indoor physical droplet behavior and global infection trends where outdoor environmental conditions are reported, it is necessary to correlate the indoor and outdoor conditions. Data from Nguyen et al. (2014) and Nguyen and Dockery (2016) were used to convert outdoor temperature and relative humidity into indoor conditions [47, 48]. Temperature and absolute humidity can directly be converted between indoors and outdoors, but there is no direct correlation for relative humidity. Thus, we first converted the absolute humidity from outdoors to indoors, and then calculated the indoor relative humidity using the converted indoor temperature.

Secondly, a cough exiting the mouth was considered to be an unsteady turbulent jet entering the ambient environment. The conditions of this environment are difficult to generalize, though. There are large seasonal and regional differences depending on HVAC operation. Referring to the ISO 7730 thermal comfort standard, this study defined an ambient air flow that is uniform in the direction of the jet exhaled from the mouth, which corresponds to the worst condition in regards to COVID-19 transmission by droplets.

An increase in relative humidity had a strong effect on curbing the spread of local cough droplets, and also resulted in a reduction of global COVID-19 transmission. However, all else being equal, temperature neither affected the spread of local cough droplets nor accounted for the global COVID-19 transmission in a statistically significant manner. Our finding is consistent with the fact that COVID-19 has been rapidly spreading both during the summer and winter [22]. Even though we found physical and empirical correlations that enhance our understanding of the spread of COVID-19, we would like to note that finer biological details were not fully considered in this study. For instance, viruses survive for a long time in both low ($< 40\%$) and high ($> 80\%$) relative humidity [49, 50]. It should also be noted that mold growth is activated at above 70% relative humidity and increases with humidity [51, 52].

In summary, we conclude that higher humidity provides a safer environment in terms of virus transmission due to physical respiratory droplet transport. However, safe RH levels should be limited to below 70%, while human discomfort at high humidity levels also needs to be considered when operating indoor HVAC systems, especially for institutions such as hospitals or schools. For future work, natural and hybrid ventilation should be considered since it can influence droplet trajectories.

## Supporting information

**S1 File. Further experiment results and correction, image processing methodology, and additional statistical analysis results.**
(DOCX)

**S2 File. Droplet evaporation video at temperature 25˚C and relative humidity 20%.**
(MP4)

## Acknowledgments

The authors gratefully acknowledge Mr. Hoonsang Lee for experimental and theoretical assistance.

## Author Contributions

**Conceptualization:** Steven Barrett, Soohyung Lee, Wontae Hwang.

**Data curation:** Jeong Suk Oh, Minhyuk Nam.

**Formal analysis:** Han June Park, Sung-Gwang Lee, Jeong Suk Oh, Minhyuk Nam.

**Funding acquisition:** Soohyung Lee, Wontae Hwang.

**Investigation:** Han June Park, Sung-Gwang Lee, Minhyuk Nam.

**Methodology:** Han June Park, Sung-Gwang Lee, Jeong Suk Oh, Minhyuk Nam.

**Project administration:** Soohyung Lee, Wontae Hwang.

**Resources:** Wontae Hwang.

**Software:** Jeong Suk Oh, Minhyuk Nam.

**Supervision:** Steven Barrett, Soohyung Lee, Wontae Hwang.

**Validation:** Han June Park, Sung-Gwang Lee.

**Visualization:** Han June Park, Sung-Gwang Lee, Jeong Suk Oh.

**Writing – original draft:** Han June Park, Sung-Gwang Lee, Minhyuk Nam, Soohyung Lee, Wontae Hwang.

**Writing – review & editing:** Steven Barrett, Soohyung Lee, Wontae Hwang.

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
