## [Decision Letter · Decision Letter 0]

7 Apr 2022

PONE-D-22-00325The effects of indoor temperature and humidity on local transmission of COVID-19 and how it relates to global trendsPLOS ONE

Dear Dr. Hwang,

Thank you for submitting your manuscript to PLOS ONE. After careful consideration, we feel that it has merit but does not fully meet PLOS ONE’s publication criteria as it currently stands. Therefore, we invite you to submit a revised version of the manuscript that addresses the points raised during the review process.

1.
Minor grammatical and spelling mistakes. Also use proper subscript and superscript2.     Address all the comments of reviewers point wise.==============================

We look forward to receiving your revised manuscript.

Kind regards,

Niaz Bahadur Khan, PhD

Academic Editor

PLOS ONE

Journal Requirements:

“The authors gratefully acknowledge the support from the Institute of Advances Machines and Design and Institute of Engineering Research at Seoul National University. Special thanks also are due to H. Lee for experimental and theoretical assistance.”

 “This work was supported by Seoul National University Research Grant in 2020 and supported under the framework of international cooperation program managed by the National Research Foundation of Korea (2020K2A9A1A01096358, FY2020).”

 “This work was supported by Seoul National University Research Grant in 2020 and supported under the framework of international cooperation program managed by the National Research Foundation of Korea (2020K2A9A1A01096358, FY2020).”

Additional Editor Comments (if provided):

1. Minor grammatical and spelling mistakes. Also use proper subscript and superscript.

2. Address all the comments of reviewers point wise.

Reviewers' comments:

Reviewer's Responses to Questions

**Comments to the Author**

1. Is the manuscript technically sound, and do the data support the conclusions?

Reviewer #1: Yes

Reviewer #2: Partly

Reviewer #3: Yes

2. Has the statistical analysis been performed appropriately and rigorously? 

Reviewer #1: Yes

Reviewer #2: Yes

Reviewer #3: Yes

3. Have the authors made all data underlying the findings in their manuscript fully available?

Reviewer #1: Yes

Reviewer #2: Yes

Reviewer #3: Yes

4. Is the manuscript presented in an intelligible fashion and written in standard English?

Reviewer #1: Yes

Reviewer #2: Yes

Reviewer #3: Yes

5. Review Comments to the Author

Reviewer #1: Study findings are interesting and paper is well written.

Statistical analysis is appropriate and rigorous.

Future works like e.g. energy analysis due to natural ventilation or hybrid ventilation are not highlighted which can be included.

Reviewer #2: Review of “The effects of indoor temperature and humidity on local transmission of COVID-19 and how it relates to global trends” (PONE-D-22-00325) by Han June Park, Sung-Gwang Lee, Jeong Suk Oh, Minhyuk Nam, Steven Barrett, Soohyung Lee, and Wontae Hwang.

This manuscript investigates the effects of temperature and relative humidity on the spread of respiratory droplets indoors. The authors hypothesize that outdoor weather data does not correlate directly to the spread of COVID-19 because most infections occur indoors; therefore, in this work, the authors determined how indoor temperature and relative humidity affect the final height and distance of evaporated droplets, and correlate the final droplet trajectory with the spread of COVID-19 for 174 countries.

This work provides a unique approach to understand the spread of respiratory viruses by studying the evaporation and transport of droplets using experiments and simulations and statistically correlating the results with the number of COVID-19 cases. The framework described in this manuscript facilitates direct comparison of outdoor weather data to indoor virus transmission, which has not been shown in the literature. However, there are some issues that need to be addressed, particularly regarding the conversion of outdoor data to indoor data. If the authors address these issues, as well as other concerns detailed in the comments below, the manuscript could be suitable for publication in PLOS One.

1. The authors start the abstract with “Over the past year…” which seems will quickly become an outdated statement and could be updated or rephrased to be more general without any implications of time.

2. (Page 4, line 81) The authors mentioned that they took a simple monthly average for the weather data, which includes only the daily average temperature. Prior work studying the seasonality of viruses have shown that the diurnal temperature range (difference between the daily maximum and minimum temperature) has a non-negligible effect on the spread of viruses, whereby high values of DTR are correlated to low infection rates (see references below). To contribute to the completeness of their study, the authors should provide additional analysis on the correlation of DTR with the spread of the virus using equation (1) in their work or discuss why DTR was not considered in the analysis.

The following references provide concepts that are relevant to the background and understanding of the importance the DTR on the spread of viruses and should be cited:

Islam A.R.M.T., Hasanuzzaman M., Azad M.A.K., Salam R., Toshi F.Z., Khan M.S.I., et al. Effect of meteorological factors on COVID-19 cases in Bangladesh. Environ Dev Sustain. 2021;23: 9139–9162. doi:10.1007/s10668-020-01016-1

Yap T.F., Decker C.J., Preston D.J. Effect of daily temperature fluctuations on virus lifetime. Science of The Total Environment. 2021;789: 148004. doi:10.1016/j.scitotenv.2021.148004

Liu J., Zhou J., Yao J., Zhang X., Li L., Xu X., et al. Impact of meteorological factors on the COVID-19 transmission: A multi-city study in China. Science of The Total Environment. 2020;726: 138513. doi:10.1016/j.scitotenv.2020.138513

3. (Page 4, line 115) The authors should define the variable Y_l,c,t here because it has not been mentioned previously.

4. (Page 4, line 115) The authors mentioned that the statistical model used here is a common method used in economics. However, the authors should still provide additional context for readers who are not familiar with economics by providing the name of the method and relevant references or include additional information in the main text or supplementary material to describe the model.

5. (Table 1) Are the results listed here averaged across all the countries and months? Is the analysis consistent across all months and locations? The authors should provide a more detailed description of their analysis process.

6. (Page 12, line 275) The authors should provide more context to justify the equation that they chose to define t_0 here.

7. (Page 12, line 278) The authors should include values of the evaporation rate in their plots or create a separate plot to illustrate the non-linear decrease.

8. (Page 12, line 278) The authors should provide the constant temperature value in the main text or in figure 2b (not only in the captions) similar to how they indicated the value of the constant relative humidity in figure 2b.

9. (Page 13, line 297) The authors can reference this work by Abkarian et al. [4] that describes the droplet transport of different speech patterns to provide insight into the droplet trajectories for not only coughing scenarios but also day-to-day conversations:

Abkarian M, Mendez S, Xue N, Yang F, Stone HA. Speech can produce jet-like transport relevant to asymptomatic spreading of virus. Proc Natl Acad Sci USA. 2020;117: 25237–25245. doi:10.1073/pnas.2012156117

10. (Page 17, line 288) The authors should clarify and define what they mean by “time elapsed”.

11. (Page 18, line 425) The authors mentioned that they are able to convert data from outdoor weather stations to indoor conditions to correlate the transmission of the virus with the trajectory of droplets (based on indoor RH and temperature), which is the foundation of their work. However, the authors do not elaborate on how they convert the outdoor data to indoor data and simply mention that the method is based on Nguyen et al., 2014 and they only included a brief discussion in their Discussion section. The work by Nguyen et al. focuses on studying the relationship between indoor and outdoor weather conditions only in Boston, Massachusetts. The authors should elaborate on how they converted their data in the main text or supplementary information to show how they extended the localized analysis in Boston to worldwide weather information.

12. (Page 21, line 476) Could the authors comment on whether the increase in RH results in a monotonic decrease in COVID-19 transmissions as mentioned in the work? Prior work has found that there exists a U-shaped dependence of the virus lifetime on relative humidity, where viruses experience a higher rate of inactivation at moderate RH, and a lower inactivation rate at low and high RH. Does the current study include cases with extreme relative humidities?

These are some references with discussion on the effects of relative humidity that could be cited:

Lin K, Marr LC. Humidity-Dependent Decay of Viruses, but Not Bacteria, in Aerosols and Droplets Follows Disinfection Kinetics. Environ Sci Technol. 2020;54: 1024–1032. doi:10.1021/acs.est.9b04959

Morris DH, Yinda KC, Gamble A, Rossine FW, Huang Q, Bushmaker T, et al. Mechanistic theory predicts the effects of temperature and humidity on inactivation of SARS-CoV-2 and other enveloped viruses. eLife. 2021;10: e65902. doi:10.7554/eLife.65902

Reviewer #3: Park et al., demonstrates the effects of indoor temperature and humidity on local

transmission of COVID-19 and how it relates to global trends. They have thoroughly examined the effect of temperature and relative humidity on respiratory droplet dynamics, and correlated this with COVID-19 transmission. They have validated the droplet evaporation model with experiments performed in an environmental chamber representing indoor conditions. Droplet trajectories were simulated, and they have analyzed how the droplet dynamics vary with temperature and humidity. They have considered two conditions. First, they used indoor conditions since most COVID-transmission is considered to occur indoors. This is the most important part of this study as the authors considered indoor transmission as primary source of COVID related infections. In order to make the connection between indoor physical droplet behavior and global infection trends where outdoor environmental conditions are reported, it is necessary to correlate the indoor and outdoor conditions. In my opinion, this is a very good study which is experimentally and model wise sound. I recommend this study to be published so that governmental bodies and health communities will be benefitted.

However, I have a few comments which needs to be addressed before the final publication.

Line 25: SARS-Cov-2 should be replaced as SARS-CoV-2.

Line 37: Again, please maintain the symmetry while writing SARS-CoV-2.

Line 66: Replace “indoors” with “indoor”.

Line 86: Write PM2.5 and PM10. Please correct this everywhere in the manuscript.

Line 114: Replace as “It may be worth explaining about the four….”.

Line 157: Why is ,,t bold in equation 1?

Line 352-359: Authors stated that when d0 is small (e.g. 70 μm), the droplet evaporates in air and becomes an aerosol for all RH cases. This is correct.

With higher RH the droplet has spread farther and the final height is lower at the fully evaporated state. This cannot be true that with high RH the aerosol has spread farther. The statement looks too general. The droplet will spread farther with low RH where the instant evaporation leads to smaller size aerosols. But, Yes, as compared to d0 of 130 μm, the aerosol of 70 μm can stay longer in air and travel farther with high humidity not as much as of low humidity. There should be clear representation here in order to avoid any misinformation.

The higher the humidity, the farther the aerosols are dispersed from the source, resulting in a lower local aerosol concentration.

This should also be corrected here.

In addition, since the evaporation rate decreases with higher RH, the influence of the gravitational body force increases accordingly, and the droplets fall further to the ground.

This is true.

That is, the final aerosol height is lower and further away from the human respiratory tract with higher humidity, resulting in a safer condition.

But the aerosol will still remain in air and still be infectious. This should not be a way to represent how higher humidity can result in safer condition. However, focus should be on because of higher chances of falling of such aerosols or droplets under high humidity conditions compared to low humidity conditions, one can say higher humidity as a safe condition. Even the 70 μm aerosols at higher humidity can not stay for longer durations as compared to a lower humidity value.

Line 484: This is true that higher humidity provides safer environment in terms of COVID protection but a limit should be mentioned here. As we know from previous studies that above 60% RH indoors, there are chances of mold growth and it increases with 80% RH. Be very careful while reporting these values.

Provide a few literature citations for safer RH values for protecting against both COVID and mold growth here.

6. PLOS authors have the option to publish the peer review history of their article (what does this mean?). If published, this will include your full peer review and any attached files.

Reviewer #1: No

Reviewer #2: No

Reviewer #3: No

---

## [Author Response · Author response to Decision Letter 0]

22 May 2022

This letter is in regards to our recent manuscript (MS #: PONE-D-22-00352), which is being revised for PLoS One, titled “The effects of indoor temperature and humidity on local transmission of COVID-19 and how it relates to global trends”.

 We thank the editor and reviewers for their precious time in reviewing our manuscript and providing valuable comments. We have carefully considered all of the comments and tried our best to address every one of them. We attached detail response letters (Word file) for each every comments.

---

## [Decision Letter · Decision Letter 1]

1 Jul 2022

PONE-D-22-00325R1The effects of indoor temperature and humidity on local transmission of COVID-19 and how it relates to global trendsPLOS ONE

Dear Dr. Hwang,

Thank you for submitting your manuscript to PLOS ONE. After careful consideration, we feel that it has merit but does not fully meet PLOS ONE’s publication criteria as it currently stands. Therefore, we invite you to submit a revised version of the manuscript that addresses the points raised during the review process. Please submit your revised manuscript by Aug 15 2022 11:59PM. If you will need more time than this to complete your revisions, please reply to this message or contact the journal office at plosone@plos.org. Please include the following items when submitting your revised manuscript:A rebuttal letter that responds to each point raised by the academic editor and reviewer(s). You should upload this letter as a separate file labeled 'Response to Reviewers'.A marked-up copy of your manuscript that highlights changes made to the original version. You should upload this as a separate file labeled 'Revised Manuscript with Track Changes'.An unmarked version of your revised paper without tracked changes. You should upload this as a separate file labeled 'Manuscript'.If applicable, we recommend that you deposit your laboratory protocols in protocols.io to enhance the reproducibility of your results. Protocols.io assigns your protocol its own identifier (DOI) so that it can be cited independently in the future. For instructions see: https://journals.plos.org/plosone/s/submission-guidelines#loc-laboratory-protocols. Additionally, PLOS ONE offers an option for publishing peer-reviewed Lab Protocol articles, which describe protocols hosted on protocols.io. Read more information on sharing protocols at https://plos.org/protocols?utm_medium=editorial-email&utm_source=authorletters&utm_campaign=protocols.

We look forward to receiving your revised manuscript.

Kind regards,

Niaz Bahadur Khan, PhD

Academic Editor

PLOS ONE

Journal Requirements:

Additional Editor Comments:

Author should carefully address the reviewer 2 comments. Addressing these minor comments will increasing the quality of the article.

Reviewers' comments:

Reviewer's Responses to Questions

**Comments to the Author**

1. If the authors have adequately addressed your comments raised in a previous round of review and you feel that this manuscript is now acceptable for publication, you may indicate that here to bypass the “Comments to the Author” section, enter your conflict of interest statement in the “Confidential to Editor” section, and submit your "Accept" recommendation.

Reviewer #2: (No Response)

Reviewer #3: (No Response)

2. Is the manuscript technically sound, and do the data support the conclusions?

Reviewer #2: Partly

Reviewer #3: Yes

3. Has the statistical analysis been performed appropriately and rigorously? 

Reviewer #2: Yes

Reviewer #3: Yes

4. Have the authors made all data underlying the findings in their manuscript fully available?

Reviewer #2: Yes

Reviewer #3: Yes

5. Is the manuscript presented in an intelligible fashion and written in standard English?

Reviewer #2: Yes

Reviewer #3: Yes

6. Review Comments to the Author

Reviewer #2: Review of “The effects of indoor temperature and humidity on local transmission of COVID-19 and how it relates to global trends” (PONE-D-22-00325) by Han June Park, Sung-Gwang Lee, Jeong Suk Oh, Minhyuk Nam, Steven Barrett, Soohyung Lee, Wontae Hwang.

This manuscript investigates the effects of temperature and relative humidity on the spread of respiratory droplets indoors. The authors hypothesize that outdoor weather data does not correlate directly to the spread of COVID-19 because most infections occur indoors; therefore, in this work, the authors determined how indoor temperature and relative humidity affect the final height and distance of evaporated droplets, and correlate the final droplet trajectory with the spread of COVID-19 for 174 countries.

This work provides a unique approach to understand the spread of respiratory viruses by studying the evaporation and transport of droplets using experiments and simulations and statistically correlating the results with the number of COVID-19 cases. The authors have addressed the majority of the comments from the previous round of review and made adequate changes to the main text to improve the clarity of the work. A minor technical issue remains regarding the dimensionless parameter, t*. If the authors satisfactorily address this issue, the manuscript will be suitable for publication in PLOS One.

1(a). In response to comment #6 in the previous round of review, the authors defined a new characteristic timescale, t_0. The authors mentioned that Figure 2 was modified to reflect this change in the variables defining t_0, but did not elaborate in their reply on how this change in variable affected the final results plotted in Figure 2.

1(b). From analyzing the dimensions of t_0 through the equation provided, the dimensions of t_0 are similar to the dimensions of the diffusion coefficient, not time:

((D_0^2)/D_g )(ρ_l/ρ_g ) = t_0

[m^2/s]^2/[m^2/s] ×[kg/m^3 ]/[kg/m^3 ] = [m^2/s]

This raises some questions about what the authors claim as a nondimensionalized parameter, t*, because time, t, has units of seconds, s, and t* is defined as t/t0. According to the equations provided, the dimensions for t* will be s2/m2. The authors should consider redefining their characteristic time or should modify their main text to reflect their definition of t*.

Reviewer #3: This is an important study showing the role of temperature and humidity on COVID-19 transmission. Authors have provided justifications to my comments. Therefore, I recommend the manuscript for publication.

7. PLOS authors have the option to publish the peer review history of their article (what does this mean?). If published, this will include your full peer review and any attached files.

Reviewer #2: No

Reviewer #3: No

---

## [Author Response · Author response to Decision Letter 1]

4 Jul 2022

We thank the editor and reviewers for their precious time in reviewing our manuscript and providing valuable comments. We have carefully considered all of the comments and tried our best to address every one of them. We attached detail response letter for every comment.

---

## [Editor Report · Decision Letter 2]

7 Jul 2022

The effects of indoor temperature and humidity on local transmission of COVID-19 and how it relates to global trends

PONE-D-22-00325R2

Dear Dr. Hwang,

We’re pleased to inform you that your manuscript has been judged scientifically suitable for publication and will be formally accepted for publication once it meets all outstanding technical requirements.

Kind regards,

Niaz Bahadur Khan, PhD

Academic Editor

PLOS ONE

Additional Editor Comments (optional):

Author addressed all the reviewers comments and the article is now suitable for publication.
---

## [Editor Report · Acceptance letter]

11 Jul 2022

PONE-D-22-00325R2 

The effects of indoor temperature and humidity on local transmission of COVID-19 and how it relates to global trends  

Dear Dr. Hwang:

I'm pleased to inform you that your manuscript has been deemed suitable for publication in PLOS ONE. Congratulations! Your manuscript is now with our production department. 

Kind regards, 

on behalf of

Dr Niaz Bahadur Khan 

Academic Editor

PLOS ONE